# Acceptability of Medical Male Circumcision as an HIV Prevention Intervention among Male Learners in a South African High School

**DOI:** 10.3390/healthcare12131350

**Published:** 2024-07-06

**Authors:** Lungani Gotye, Sibusiso C. Nomatshila, Kedibone Maake, Wezile Chitha, Sikhumbuzo A. Mabunda, Anam Nyembezi

**Affiliations:** 1Department of Public Health, Walter Sisulu University, Mthatha 5117, South Africa; lungani@neom-hi.org (L.G.); snomatshila@wsu.ac.za (S.C.N.); wchitha@wsu.ac.za (W.C.); s.mabunda@unsw.edu.au (S.A.M.); anyembezi@uwc.ac.za (A.N.); 2School of Population Health, University of New South Wales, Sydney, NSW 2052, Australia; 3George Institute for Global Health, University of New South Wales, Sydney, NSW 2052, Australia

**Keywords:** AmaXhosa, adolescent males, traditional male circumcision (TMC), voluntary male medical circumcision (VMMC)

## Abstract

Circumcision is a long-standing and frequently performed surgical procedure which holds significant cultural significance among AmaXhosa people in South Africa. Due to cultural reasons in some parts of Africa, the integration of medical male circumcision with traditional manhood initiation rituals still lacks acceptance. This study examined the level of knowledge and acceptance of voluntary male medical circumcision (VMMC) among young males in a selected high school in the Nyandeni District of the Eastern Cape Province, South Africa. A descriptive, quantitative, cross-sectional study was conducted, and a self-administered questionnaire was used to obtain information on sociodemographic characteristics, knowledge of VMMC, perceptions of VMMC, and circumcision practices. One hundred participants were recruited from both grades 11 and 12, and 82% of the participants indicated that they had received information about VMMC. Most of the respondents (88%) preferred traditional male circumcision (TMC), and only 12% of respondents preferred VMMC. The participants displayed a good understanding of the distinction between VMMC and TMC. However, results from the study showed that most respondents exhibited low acceptability and knowledge towards VMMC. These findings highlight the need to develop evidence-based strategies to enhance learners’ knowledge and acceptance of VMMC.

## 1. Introduction

Circumcision is one of the oldest and most common surgical procedures [1,2,3]. In erstwhile times, religious, covenant establishment (example of Genesis 17:10 in the Bible), medical, initiation ritual, and a rite of passage into manhood were major indicators for circumcision [4,5,6]. These indicators defined the place and person to conduct the procedure and most often the outcomes could be positive (survival) or detrimental (death) to the individual being circumcised.

Among the AmaXhosa, in South Africa in particular, circumcision is of paramount importance and highly valued by every young man, because it symbolises communal pride and individual worth [7]. Nearly 27 million voluntary medical male circumcision (VMMC) had been performed in 15 priority countries by the end of 2019, and more than 60% of the 2020 VMMC target had been reached before services were disrupted by the SARS-CoV-2 pandemic [8].

With advancement in biomedical science and changing health transition of populations aligned with current pandemics such as HIV/AIDS, male circumcision is not only an indication for religious, cultural, or medical fulfilment, but is now considered as a public health measure to reduce the incidence of HIV infection and the overall prevalence rate of this disease among populations [4,5,9].

The World health Organisation (WHO) and UNAIDS organised a consultation of modelers and policymakers to assess various models and projections and develop key messages to guide strategic directions over the following five years, and this was done in anticipation of a new phase of VMMC interventions, which was expected to last until 2021 [10]. There has been a significant investment in implementing VMMC programmes for HIV prevention in Southern and Eastern Africa since the WHO/UNAIDS recommended that medical male circumcision be considered an additional method of HIV prevention in 2007 and for it to be rapidly scaled up in countries with low prevalence of circumcision and high prevalence of HIV [10]. About 11.7 million male adolescents and adults have been circumcised by the end of 2015 [10].

Other publications predicted that an extensive practice of VMMC in sub-Saharan Africa would prevent almost six million HIV infections over 20 years, as would three million deaths among both men and women [11]. Three randomised controlled trials (RTCs) conducted in different African countries involving predominantly young, HIV-negative adult males found conclusive evidence that male circumcision has a protective effect against HIV infection in men [12]. The trials, which took place in Orange Farm (South Africa), Kisumu (Kenya), and the rural Rakai Province of Uganda, showed that circumcision reduces the risk of HIV acquisition by approximately 50–70%, and due to these clear results, the authors reported that it was considered unethical not to offer circumcision to the control group before the planned end date of the trials [12].

In certain African regions, combining medical male circumcision with traditional manhood initiation ceremonies faces challenges due to cultural reasons; Wilcken et al. (2010) reported that 70% of men express fear of stigma if they undergo medical circumcision; additionally, initiates may hesitate to seek post-circumcision care at medical facilities because leaving the initiation school is considered shameful in certain contexts [13]. For example, in Libode village of Nyandeni sub-district in the Eastern Cape Province, South Africa, VMMC is said to be culturally unacceptable [14]. This is because male circumcision is regarded as a cultural practice that serves as a transition into manhood [14]. In some communities, respected authorities like traditional leaders argue that VMMC is undermining their culture and customs [15]. Twenty-eight traditional initiates admitted to a South African district hospital in the Eastern Cape province after complications of their traditional male circumcision (TMC) previously stated that “a real man does not use western medicine but traditional medicine, therefore traditional male circumcision (TMC) is number one” [16].

The main aim of the study was to examine the level of knowledge and acceptability of VMMC among young males in a selected high school in the rural Eastern Cape in South Africa. It was conducted to identify sources of information and identifying socio-demographic factors that influence the acceptability of VMMC as an HIV prevention intervention among Grade 11 and 12 males aged 15 to 20 years. The study was conducted to give a better understanding about VMMC and the benefits after circumcision as compared to TMC which is mostly performed by non-health professionals and lay members of the community. Even though there are no complications in most initiates, these complications often lead to hospital admissions, penile amputations, and/or death of those affected [13].

The results of this study are intended to be a useful reference point for educational purposes for the acceptance of VMMC, and to circumcise safely whilst respecting the practice of culture. The findings and recommendations will help to put mechanisms in place to curb the high number of admissions to hospitals due to septic circumcision, dehydration, gangrene, amputations, and death related to TMC. This study also intended to make a significant impact on the life of people in the study population and other similar populations.

The findings and recommendations of the study may influence policy and decision makers, government, NGOs, and communities. Findings of this study are also intending to improve the sexual reproductive health of young males seeking circumcision. Results of this research will contribute to the body of scientific knowledge around circumcision. A copy of this research will be made available to the study site and the neighbouring schools and communities. Lastly, the study will also help the families, communities, and society at large to avoid complications resulting from TMC that may lead to loss of life by referring their children to health facilities for VMMC before going to the ‘*mountain*’ or TMC site for manhood laws.

## 2. Materials and Methods

Study design. A descriptive quantitative cross-sectional study was used to answer the research question and objectives. This design was chosen because it was a cost-effective way to answer the research question in a snapshot without the need for a follow-up. The study was conducted on the 23 March 2018.

Study setting. Eastern Cape Province is South Africa’s second largest province by surface area and fourth most populous [17]. The largest district, OR Tambo, is one of eight districts in the province, and it has five sub-districts [17]. The Eastern Cape province’s African Black population is almost homogenously made of AmaXhosa, and there is an expectation for all boys who reach the age of 18 to undergo TMC. It is the only South African province where there is an expectation for all African Black males to undergo this practice in all households. This was therefore the reason for the choice of this site. The study was conducted at a conveniently selected high school located in the small town of Libode in the Nyandeni sub-district.

Study population. The study population comprised all (n = 199) the Grade 11 and 12 males aged between 15 and 20 years regardless of their circumcision status. Absence from school and refusal to consent were the only exclusion criteria.

Sampling and Sample size. The required sample size was determined using the equation for a one-sided 95% Confidence Interval for a cross-sectional study, $n=\frac{p\left(100-p\right)z^{2}}{d^{2}}$, where z = 1.96. The proportion of acceptability of male circumcision (p) was estimated to be 50% since there was no literature guidance; and the desired precision (d) was set at 10%. This then yielded a minimum sample size of 96, which was rounded off to 100. All males in Grades 11 and 12 were assigned random numbers, and 100 numbers were randomly drawn with the assistance of Microsoft Excel Office 2016. Each drawn number was called, and individuals were privately offered an opportunity to consent or refuse participation. All drawn 100 participants were accepted. As a result, 57 Grade 12s and 43 Grade 11s participants were sampled, respectively.

Data collection. The questionnaire (Appendix A) is a self-administered instrument that sought to find information on socio-demographic characteristics, knowledge of VMMC, perceptions on VMMC, and circumcision practices. The questionnaire was developed in English and IsiXhosa (the local language) to allow for participants to choose a preferred language. A pilot study was conducted among twenty male learners in a different school within the same district to test the reliability of the questionnaire, identify its appropriateness, and measure its length. In addition, two questions were repeated rephrased in different parts of the questionnaire to ascertain whether the same response was given. A set of questions were developed and submitted to two public health professionals to ascertain content and context validity of the instrument. Participants were not trained on how to use the instrument, but it was explained to them and circulated for answering. The use of a self-administered questionnaire and monitoring of participants as they completed their questionnaires minimised the possibility of social desirability bias.

Data analysis and management. Categorical data are presented using frequencies, percentages, and graphs. Association between two categorical variables was conducted using the chi-squared statistic or the Fisher’s exact test if the expected frequencies were ≤5. Data were analysed using Statistical Package for Social Sciences (SPSS) version 28. Knowledge was determined through the post hoc analysis of fourteen questions where a correct response was allocated a score of ‘1’ and an incorrect response a score of ‘0’. A percentage was then determined, and participants who scored above 50% were categorised as having obtained an adequate score and those below 50% as having obtained a poor score. Logistic regression analyses were performed to compute bivariate associations of purposefully selected variables [18] associated with the acceptability of VMMC. The Odds Ratio (OR) is the measure of association used. The 95% Confidence Interval (95% CI) was used to estimate the precision of estimates. The *p*-value for statistical significance was set at *p*-value = 0.05.

## 3. Results

### 3.1. Socio-Demographic Profile of the Sample Population

One hundred (100) participants were recruited from both Grade 11 and Grade 12. The median age was 19 years. Most of the participants (57%) were in Grade 12, Christian (72%), resided in rural areas (79%), and their parents were married/living together (43%). Forty-one (41%) percent reported that their fathers were formally employed, while (40%) reported that their mothers were formally employed. Table 1 is the description of the participants’ socio-demographic characteristics.

### 3.2. Source of VMMC Information among the Participants

Respondents were asked to indicate whether they had ever come across any information on VMMC. Most of the participants (82%) indicated that they had received information on VMMC.

Participants were asked on their preferred source of information, and they could list more than one source.

Most of the respondents (58% or n = 58) cited television as the main source of information for VMMC, followed by radio (55%), clinic or hospital (32%), newspaper (32%), community events (31%), friends or relatives (28%), and Facebook or Twitter (26%). Three participants (3%) could not recall their source of information.

### 3.3. Knowledge about VMMC

The suggestion that VMMC entirely prevents HIV infection among females was high among participants in Grade 11 (64.3% or n = 9) (Table 2).

The average knowledge score was 39%, with a minimum of 0%, and the highest score was 78.6% (standard deviation = 18.1%). A comparison of knowledge about VMMC between Grade 11 and Grade 12 shows no statistical differences (Table 3) even though those with adequate knowledge were 1.6 times more likely to be in Grade 11 (OR = 1.6; 95% CI = 0.7–3.6; *p*-value = 0.290).

### 3.4. Knowledge on Complications of VMMC

Among the study sample, 56% indicated that there were complications related to VMMC (Figure 1).

The results presented in Table 4 show the different complications related to VMMC, and participants could indicate more than one complication.

Among those who reported that there were complications related to VMMC, most of the participants (46.4%) identified infections (urinary tract infection), followed by “severe pains of the penis” (30.4%) and “wound not healing at 60 days after surgery” (30.4%). Other complications of VMMC identified were “continuous excessive bleeding” (26.8%) and partial penile amputation (16.1%).

### 3.5. Acceptability of VMMC as an HIV Prevention Intervention

Table 5 shows the level of acceptability of VMMC as an HIV prevention intervention. For personal acceptance, only a minority of the participants (31 participants) would consider undergoing VMMC. Again, 28 and 24 participants suggested that either their parents or their family members would, respectively, allow them to undergo VMMC. Among the participants, only 14% of their male friends or 27% of their female friends would allow them to undergo VMMC. Respondents thought that only 16% of the participants’ male schoolmates or 21% of the participants’ female schoolmates would allow them to undergo VMMC. Furthermore, only 14% of the participants suggested that their community members would allow them to undergo VMMC.

### 3.6. Preferential Type of Circumcision

The type of circumcision preferred was assessed at individual and referral levels. The findings obtained are presented in Figure 2 and Figure 3.

Only 27% of the respondents would refer a friend for VMMC and the rest (73%) would not (Figure 2).

In addition, only 25% of the respondents would refer a relative for VMMC and 75% would not (Figure 2).

Compared to VMMC, most respondents (88%) preferred TMC (Figure 3).

### 3.7. Factors Influencing the Acceptability of VMMC

Factors influencing the acceptability of VMMC were assessed and are shown in Table 6. Acceptability of VMMC was significantly associated with the following factors: the suggestion that VMMC reduced the risk of HIV infection among males (*p* = 0.026) and among females (*p* = 0.012); personal acceptance to undergo VMMC (*p* = 0.001); acceptability of parents (*p* = 0.034) and family members (*p* = 0.005); acceptability of female friends (*p* = 0.015) and female schoolmates (*p* = 0.017); and accepting to refer a friend (*p* < 0.001) and a relative (*p* < 0.001).

The acceptability of VMMC was not statistically associated with any socio-demographic characteristics (Table 7).

### 3.8. Independent Factors Associated with Acceptability of VMMC

Bivariate associations of selected factors influencing the acceptability of VMMC (Table 8).

Table 8 presents the bivariate associations of VMMC. Those participants who would consider undergoing VMMC were 16 times more likely to accept VMMC, and this was statistically significant (OR= 16.0; *p* = 0.001). Other participants who were significantly likely to accept VMMC are those who would refer a relative for VMMC (OR = 13.5; *p*-value < 0.001), those who thought VMMC reduced the risk of HIV among females (OR = 5.1; *p*-value = 0.011), and those who thought that female friends would not discourage VMMC (OR = 4.8; *p*-value = 0.015).

## 4. Discussion

Male circumcision is associated with a reduction in risk of HIV/AIDS and STI transmission among males [19]. VMMC has been reported to be much safer than TMC as it is associated with fewer complications that could range from local sepsis and amputation of penis to more global complications and even death. This study provides a good basis for large-scale studies on the understanding of socio-demographic factors that affect uptake of VMMC and can be used to generate hypotheses for future studies. In addition, more directed and structured interventions can be used as this study also explores the best medium of communication or marketing for the population.

This study is unique in that it does not only address the research paucity in this area but has comprehensively sought to explore the perceptions, practices, and knowledge of VMMC among senior high school learners in a high HIV burden rural population that practices TMC as a cultural norm. The population did not have adequate knowledge about VMMC, did not accept it, would not recommend it to a friend or relative, and mostly relied on television and radio for information. Educational strategies should emphasise that VMMC is not there to replace traditional practices but rather to supplement cultural practices by performing the surgical procedure in a sterile and safe environment whilst living the traditional ritual of initiation untouched.

The age group and current school Grade of the respondents suggest that they began school at around the age of seven years in accordance with the South African School Act of 1996 [20]. Further consistent with the literature, which describes a community that has a similar setting as the study sight as a highly homogenous community, is the fact that almost 72% of the population were Christian [21]. The higher proportion of individuals who did not know their father’s jobs (23%) compared to those who did not know their mother’s job’s (5%) could suggest a high proportion of learners who lived with only their mother as a primary caregiver. This is also consistent with the statistics that South Africa reports for the area which reported that almost 57.6% of females headed households in this study area [17].

Radio and television were the most cited sources of information on VMMC as an HIV prevention intervention at 55% and 58%, respectively. This resonates with findings by Spyrellis et al. [11], Hatzold [22], and Chikutsa [23] who all previously reported that the primary source of information about VMMC in South Africa is the radio and television. Evidence-based approaches should be used to develop health promotion strategies and marketing using these media platforms.

Sources through which information could be obtained by face-to-face interactive sessions such as in the clinics, hospitals (due to lack of visibility of community healthcare workers), HIV/AIDS organisations or community events were cited by a minority of respondents. These findings denote that there is a high level of awareness about VMMC but low levels of education and empowerment, which are the pillars of acceptability.

The present study established that the level of knowledge about VMMC was low. This could be indicative of lack of education and empowerment communication strategies. This is because of the sources through which they obtained information concerning VMMC.

Effective strategies for improving knowledge are integrative and should monitor the competencies of the individuals who deliver the information on TV or Radio. Such strategies should also place emphasis on the quality of the VMMC content that is communicated to the public on all forms of media and public platforms and increase face-to-face sources of reliable information (e.g., community health workers, health promoters, school health nurses, etc.).

Bertrand [24] and WHO [25] reported that demand creation for mobilisation and motivation of men to access VMMC services should use targeted and age-specific communication strategies. Therefore, all communication strategies be it for awareness or educational purposes should provide clear information on the procedure, pre-test counselling, interpersonal attributes, and cost of the procedure [26,27]. The findings obtained from the present study could signify that the communication strategies that have been used to speed up the uptake of VMMC failed to provide accurate and adequate information.

To confirm the above, findings from this study show that the respondents knew that VMMC was different from TMC, that VMMC is safer when it is performed according to protocol, and that there were complications related to VMMC. However, most of them were not knowledgeable about who performs the procedure and where the procedure was peformed. The findings also showed that the respondents had low levels of knowledge about VMMC as an HIV prevention intervention and when naming complications related to VMMC. These findings further show a lack of trustworthy information about VMMC.

Most participants (88%) preferred TMC to VMMC. These participants who preferred TMC supposedly due to social expectation for every male to go through the circumcision process the traditional way. This low level of VMMC acceptability could be because of the cultural and social ties of TMC. It is believed that AmaXhosa men who are not circumcised traditionally cannot be accepted as men [28]. The respondents noted that they would not undergo VMMC. They would also not recommend a friend or relative to undergo VMMC [28]. Peltzer et al. [28] also noted that rejection of VMMC is influenced by the number of factors such as equipment used before and after the procedure, as well as the overall context for and meaning of the surgery (for HIV prevention and health, compared with a rite of passage to manhood). Participants of this study also confirmed that their parents, family members, peers (male friends, female friends, male school mates, female school mates), and community members would not allow them to undergo VMMC as an HIV prevention intervention. The low level of acceptability could also be because of the gap in knowledge on the health benefits of VMMC.

These findings contradict those reported by Peltzer and Mlambo and Milford [29,30]. They documented a high level of acceptability of VMMC among Black African and Coloured population from both rural and urban areas from nine provinces in South Africa [29,30]. The reasons for the contradictions could be due to the fact that participants in Peltzer et al.’s article were not stratified by province and were not necessarily high school learners.

Furthermore, the evidence discovered from independent factors associated with acceptability of VMMC revealed that VMMC is gaining less acceptance as there are only few participants (10/31) who agree to undergo VMMC and 2/69 did not agree. Westercamp and Bailey [31] recorded quite a few factors that influence the non-acceptability of VMMC such as pain, culture and religion, complications, and adverse effects. Some participants of the study by Ngalande et al. [32] recognised the need to pay for circumcision services because a free circumcision was viewed as being of potentially poor quality.

Despite efforts to minimise them, this study had some limitations. Participants were not asked to state reasons why they preferred the type of circumcision they chose over the other. However, this will be a subject of future studies as the objective of this study was to merely determine acceptability of VMMC. The circumcision status of the respondent was not determined. Evidence of whether these respondents have been circumcised (TMC or VMMC) or not could possibly provide added knowledge on the extent of the problem. It could have been great to classify the respondents as circumcised or uncircumcised, and if circumcised, they were stratified by the type of circumcision. As this was a quantitative study, it was not possible to go in-depth into the reasons for the low level of knowledge and poor acceptability. The study sample was extracted from one rural high school and two grades; therefore, findings cannot be generalised to all other rural males in the area. Notwithstanding, this study has been able to ascertain that learners in this high school have different levels of knowledge and acceptability. Future research needs to be undertaken to explore if this issue is not widespread and how the knowledge gaps could be closed.

## 5. Conclusions

Respondents demonstrated that they had a good understanding that VMMC is different from TMC. Though they had heard about VMMC, they had poor knowledge of how VMMC is performed, who performs VMMC, and the complications of VMMC. The major sources through which they had access to information seem to lack the education and empowerment elements that are needed to effectively educate people and enable them to take up VMMC as the preferred method for circumcision.

Most respondents demonstrated low acceptability and knowledge of VMMC. Only a small percentage of respondents had a positive attitude towards VMMC. Evidence-based strategies need to be established to improve learners’ knowledge and improve acceptability of VMMC.

## Figures and Tables

**Figure 1 healthcare-12-01350-f001:**
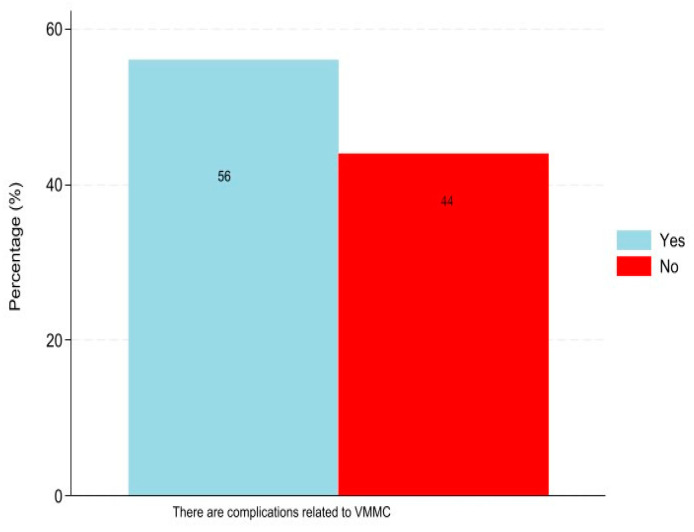
Responses (%) on whether there are complications with VMMC.

**Figure 2 healthcare-12-01350-f002:**
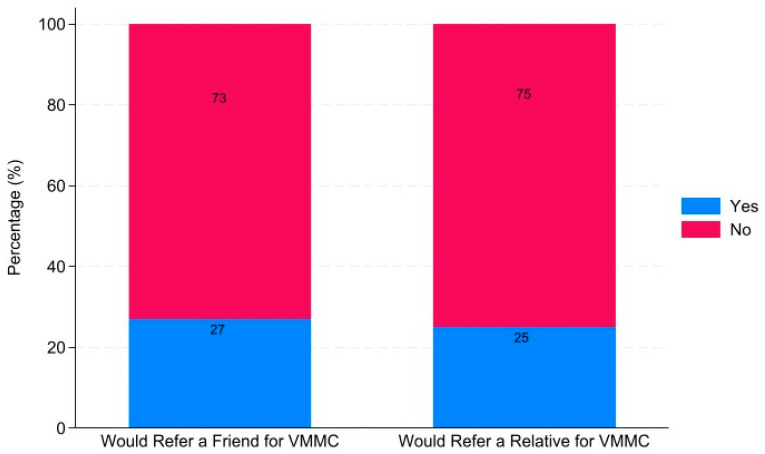
Bar graph showing the proportion of participants who would refer a friend or relative for VMMC.

**Figure 3 healthcare-12-01350-f003:**
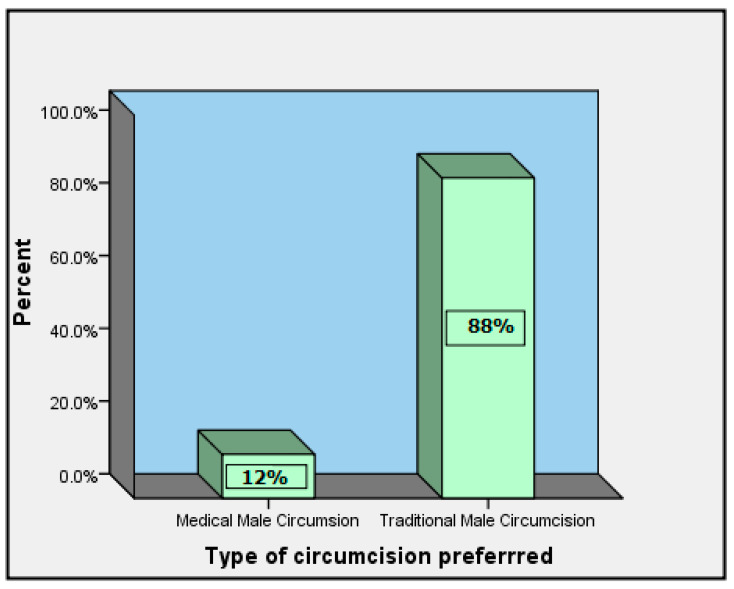
Preferential type of circumcision.

**Table 1 healthcare-12-01350-t001:** Socio-demographic characteristics.

Characteristics	Frequency	Frequency (N = 100)	Percentage (%)
Grade; n or %	Grade 11	43	43
	Grade 12	57	57
Age, years; n or %	15 years	2	2
	16 years	3	3
	17 years	9	9
	18 years	27	27
	19 years	31	31
	20 years	28	28
Religion; n or %	Christian	72	72
	Rastafarian	8	8
	Hindu	1	1
	Other	3	3
	No religion	16	16
Place of residence; n or %	Rural area	79	79
	Urban area	16	16
	Informal settlement	5	5
Marital status of parents; n or %	Married or cohabiting	43	43
	Separated	8	8
	Divorced	3	3
	Widowed	3	3
	Single	43	43
Employment status of father; n or %	Formally employed	41	41
	Self-employed	24	24
	Don’t know	23	23
	Father deceased	12	12
Employment status of mother; n or %	Formally employed	40	40
	Self-employed	40	40
	Don’t know	5	5
	Mother deceased	15	15

**Table 2 healthcare-12-01350-t002:** Knowledge of VMMC and its advantages and differences with TMC.

Variable (N = 100)	Yes	No	I Don’t Know
	n/%	n/%	n/%
VMMC is different to TMC	93	1	6
VMMC is safe	57	5	38
VMMC is done at the hospital	55	11	34
VMMC is done by a tradition surgeon	41	40	19
VMMC is done by a doctor	38	16	46
VMMC reduces the risk of HIV among males	36	15	49
VMMC reduces the risk of STI among males	35	13	52
VMMC is done at the clinic	34	32	34
VMMC entirely prevent HIV infection among males	27	21	52
VMMC reduces the risk of STI among females	26	16	58
VMMC reduces the risk of HIV among females	26	23	51
VMMC is done by any circumcised men	25	31	44
VMMC entirely prevent HIV infection among females	14	26	60
VMMC is done by a nurse	13	31	56

**Table 3 healthcare-12-01350-t003:** Comparison of knowledge about VMMC between Grade 11 and 12 participants.

Grade	Adequate Knowledge	Poor Knowledge	Total	OR (95% CI)	*p*-Value
	n (%)	n (%)	n (%)		
Grade 11	18 (41.9)	25 (58.1)	43 (100.0)	1	Reference
Grade 12	18 (31.6)	39 (68.4)	57 (100.0)	1.6 (0.7–3.6)	0.290
Total	36 (36.0)	64 (64.0)	100 (100.0)		

OR = Odds Ratio; 95% CI = 95% Confidence Interval.

**Table 4 healthcare-12-01350-t004:** Complications related to VMMC (n = 56).

Variable	Frequency (n)	Percentage (%)
Infections (urinary tract infection)	26	46.4
Severe pains of the penis	17	30.4
Wound not healing at 60 days after surgery	17	30.4
Continuous excessive bleeding	15	26.8
Partial amputation	9	16.1

**Table 5 healthcare-12-01350-t005:** Level of acceptability of VMMC as an HIV prevention intervention.

Variable	Agree	Unsure	Disagree
	n	n	n
I would consider undergoing VMMC	31	27	42
Parents would allow me to undergo VMMC	28	23	49
Family members would allow me to undergo VMMC	24	26	51
Male friends would allow me to undergo VMMC	14	28	58
Female friends would allow me to undergo VMMC	27	27	46
Male schoolmates would allow me to undergo VMMC	16	27	57
Female schoolmates would allow me to undergo VMMC	21	31	48
Community members would allow me to undergo VMMC	14	36	50

**Table 6 healthcare-12-01350-t006:** Factors influencing acceptability of VMMC.

VMMC	VMMC		TMC		*p*-Value
	n	(%)	n	%	
	12		88		
Performed by a nurse; n (%)					
Yes	4	(30.8)	9	(69.2)	0.048
No	8	(9.2)	79	(90.8)	
Reduces HIV risk among males; n (%)					
Yes	8	(22.2)	28	(77.8)	0.026
No	4	(6.3)	60	(93.8)	
Reduces HIV risk among females; n (%)					
Yes	7	(26.9)	19	(73.1)	0.012
No	5	(6.8)	69	(93.2)	
Information at clinic or hospital; n (%)					
Yes	7	(22.6)	24	(77.4)	0.044
No	5	(7.3)	64	(92.8)	
I would consider undergoing VMMC; n (%)					
Yes	10	(32.3)	21	(67.7)	<0.001
No	2	(2.9)	67	(97.1)	
Parents won’t discourage VMMC; n (%)					
Yes	7	(25.0)	21	(75.0)	0.034
No	5	(6.9)	67	(93.1)	
Family members won’t discourage VMMC; n (%)					
Yes	7	(30.4)	16	(69.6)	0.005
No	5	(6.5)	72	(93.5)	
Male friends won’t discourage VMMC; n (%)					
Yes	5	(38.5)	8	(61.5)	0.008
No	7	(8.1)	80	(92.0)	
Female friends won’t discourage VMMC; n (%)					
Yes	7	(25.9)	20	(74.1)	0.015
No	5	(6.9)	68	(93.2)	
Male schoolmates won’t discourage VMMC; n (%)					
Yes	3	(18.8)	13	(81.3)	0.402
No	9	(10.7)	75	(89.3)	
Female schoolmates won’t discourage VMMC; n (%)					
Yes	6	(28.6)	15	(71.4)	0.017
No	6	(7.6)	73	(92.4)	
The community won’t discourage VMMC; n (%)					
Yes	5	(35.7)	9	(64.3)	0.012
No	7	(8.1)	79	(91.9)	
I would refer a friend for VMMC; n (%)					
Yes	9	(33.3)	18	(66.7)	<0.001
No	3	(4.1)	70	(95.9)	
I would refer a relative for VMMC; n (%)					
Yes	9	(36.0)	16	(64.0)	<0.001
No	3	(4.0)	72	(96.0)	

**Table 7 healthcare-12-01350-t007:** Effects of demographic factors on acceptability of VMMC.

Demographic Variables	VMMC		TMC		*p*-Value
	n	(%)	n	(%)	
	12		88		
Grade attending; n (%)					
11	5	(11.6)	38	(88.4)	0.921 *
12	7	(12.3)	50	(87.7)	
Religion; n (%)					
Christian	9	(12.5)	63	(87.5)	0.699
Rastafarian	0	(0)	1	(100.0)	
Hindu	0	(0)	8	(100.0)	
No religion	3	(18.8)	13	(81.3)	
Other	0	(0)	3	(100.0)	
Place of residence; n (%)					
Rural area	9	(11.4)	14	(87.5)	0.698
Urban area	2	(12.5)	70	(88.6)	
Informal settlement	1	(20.0)	4	(80.0)	
Marital status of parents; n (%)					
Married/living together	7	(16.3)	36	(83.7)	0.875
Single	4	(9.3)	39	(90.7)	
Separated	1	(12.5)	7	(87.5)	
Divorced	0	(0)	3	(100.0)	
Widowed	0	(0)	3	(100.0)	
Employment status of father; n (%)					
Formally employed	6	(14.6)	35	(85.4)	0.599
Self-employed	1	(4.2)	23	(95.8)	
I don’t know about my father	3	(13.0)	20	(87.0)	
Father deceased	2	(16.7)	10	(83.3)	
Employment status of mother; n (%)					
Formally employed	4	(10.0)	36	(90.0)	0.885
Self-employed	6	(15.0)	34	(85.0)	
I don’t know about my mother	0	(0)	5	(100.0)	
Mother deceased	2	(13.3)	13	(86.7)	

* The Chi-squared statistics was used.

**Table 8 healthcare-12-01350-t008:** Independent factors associated with acceptability of VMMC.

Characteristic	n/N	OR	95% CI	*p*-Value
Reduces HIV risk among females				
No	5/74	ref		ref
Yes	7/26	5.1	(1.4–17.8)	0.011
Would consider undergoing VMMC				
Do not agree	2/69	ref		ref
Agree	10/31	16.0	(3.2–78.6)	0.001
Female friends won’t discourage VMMC				
Do not agree	5/73	ref		ref
Agree	7/27	4.8	(1.4–16.6)	0.015
Would refer a relative for VMMC				
No	3/75	ref		ref
Yes	9/25	13.5	(3.3–55.5)	<0.001

OR = Odds Ratio. ref = reference. 95% CI = 95% Confidence Interval.

## Data Availability

All data reported are available from L.G. upon reasonable request.

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
