# Peer review of "Acceptability of Medical Male Circumcision as an HIV Prevention Intervention among Male Learners in a South African High School"

_healthcare, 2024, doi:10.3390/healthcare12131350_

Round 1

Reviewer 1 Report

Comments and Suggestions for Authors

Thank you for the opportunity to review this manuscript. The introduction orients the reader on the topic. The topic is of relevance to healthcare workers and policymakers. The materials and methods section need to be strengthened and some of the analysis needs to be changed or improved. The results section can also need improvement.

MAJOR REVISIONS

Materials and Methods

1. The authors should provide more information on why this study setting was chosen for the study.

2. In data collection, the authors have to state what the reliability of the questionnaire was and what method was used to determine the reliability.

3. In data analysis:

·         The authors state in lines 163-165, ‘Logistic regression analyses were performed to compute bivariate associations of purposefully selected variables associated with the acceptability of VMMC.’ Can the authors justify selecting variables purposively. Usually, all the variables that are significant in Chi-square tests are included in logistic regression. I suggest they do this and provide all the results of the significant variables taken from Chi-square tests in their logistic regression results.

·         Instead of looking at individual questions to determine the knowledge level of VMMC, the authors should have scored the responses to each question, added them to get total scores, and then divide these scores into good knowledge level and poor knowledge level. Such an analysis would be more informative because, as it is, they cannot comment on whether the participants had good knowledge of VMMC or not. I suggest the authors re-analyze their data by dividing knowledge levels into good and bad.

Results

5. The authors should ensure that all the results show absolute numbers and percentages when reporting frequencies, for example, in line 172 they state, ‘Most of the participants (57%) were in grade 12, …’ This should be ‘Most of the participants (n=57; 57%) were in grade 12, …’

6. In section 3.7, the authors provide Table 6 which shows factors influencing acceptability of VMMC using Chi-square tests. Why did they not perform logistic regression on the significant factors to determine the extent of the associations? I suggest that this be performed to strengthen the results of the study.

Discussion

7. In lines 318-330, authors provide a discussion on the sociodemographic characteristics of participants. This discussion should only include those characteristics that have an impact on the study findings.

8. In line 345, the authors state, ‘The present study established that the level of knowledge about VMMC was low.’ How was the level of knowledge determined? Please refer to comment 2.

9. In lines 388-391, the authors reported on a study that gave contradicting results. Can they explain the possible reasons for these contradictions.

Conclusion

10. In line 425-428, the authors state, ‘Socio-economic characteristics that influence acceptability of VMMC was the employment status of mother.’ However, Table 7 does not show any associations. Where did they get this?

MINOR REVISIONS

11. Keywords should be in alphabetic order and should include ‘adolescent males’

12. In line 107, the authors state, ‘The findings and recommendations of the study will influence policy and decision…’ Is this absolute? Rather, use ‘may’ instead of ‘will’ where it is not definite throughout this paragraph.

13. In line 163, state the version of SPSS used.

Comments on the Quality of English Language

Minor editing required.

Reviewer 2 Report

Comments and Suggestions for Authors

Acceptability of Medical Male Circumcision as an HIV prevention intervention among male learners in a South African high school.

The authors have carried out a very interesting and original work, in which they assess the importance of the medical male circumcision (VVMC), on VIH prevention, and its advantage with traditional male circumcision (TMC).

The publication described the state of the art of the study problem, and the section sing literature which closely related to the study topic.

The article is scientifically exciting; however, it has some minor aspects that need to be corrected.

Introduction.

The introduction described the state of the art of study issue.

Materials and Methods.

The experiment of this study is well-designed.

The authors should omit the map of the area under study, since it is already described in the text.

Results and Discussion.

The results and discussion are described the findings of the manuscript, and the section singing current literature that are closely related to the study topic.

Lines 318-323 should be included in results.

The authors should report only the most significant results and not repeat the data contained in the table or in the figure.

Lines 183-185: Lines 192 -197: Repeat the results shown in Figure 1.

Lines 192 -197: Repeat the results shown in Figure 2.

The authors should omit Figures 1 and 2.

Figures 4 and 5 should be unified.

Authors should include the signification in some tables, such as 2,4,5,

Line 388: these findings contradict those reported by

References

Most of the references listed in the manuscript are not recent. There are too many references from more than 5 years of publications. Some are necessary, but others need to be updated.

Round 2

Reviewer 1 Report

Comments and Suggestions for Authors

The authors have adequately addressed all my comments.